# XFlow: Cross-modal Dataflow Neural Networks for Audiovisual Classification

**Cătălina Cangea, Petar Veličković & Pietro Liò** [*]
Department of Computer Science and Technology
University of Cambridge
Cambridge, CB3 0FD, United Kingdom
`{Catalina.Cangea,Petar.Velickovic,Pietro.Lio}@cst.cam.ac.uk`

## Abstract

We propose two *multimodal deep learning architectures* that allow for cross-modal dataflow (XFlow) between several feature extractors, deriving more interpretable features and obtaining a better representation than through unimodal learning. These models can usefully exploit correlations between audio and visual data, which have a different dimensionality and are *nontrivially exchangeable*. Our work improves on existing multimodal research in two essential ways: (1) it presents a novel method for performing cross-modality (which could easily be generalised to other kinds of data) and (2) extends the previously proposed *cross-connections* which only transfer information between streams that process *compatible* data. We also *illustrate some of the representations* learned by the connections and present *Digits*, a new dataset consisting of three audiovisual data types. Both architectures outperformed their baselines and achieved state-of-the-art results on *AVletters* and *CUAVE*.

## 1 Introduction

An interesting extension of unimodal learning consists of deep models which "fuse" several modalities (for example, sound, image or text) and learn a shared representation, outperforming previous architectures on discriminative tasks. However, the cross-modality in existing models— (Ngiam et al., 2011), (Srivastava & Salakhutdinov, 2012) and (Aytar et al., 2017)—only occurs after features are learned. This prevents the unimodal feature extractors from exploiting any information contained within other modalities.

Our work has focused on facilitating this information exchange, which poses a highly nontrivial problem as it takes place between data of varying dimensionality (for example, 1D/2D for audiovisual data). We have generalised cross-connections (Veličković et al., 2016) to exploit the correlations between audio and image; our models have surpassed current state-of-the-art results. We also introduce *Digits*—a novel, open dataset of superior quality to other existing benchmark audiovisual datasets [1]. Finally, we analyse the representations learned by cross-connections, deriving useful conclusions about their mutual constructiveness for the classification task—a step towards addressing the "black box" problem encountered in deep learning.

## 2 XFlow Models

### 2.1 CNN × MLP, {CNN × MLP}–LSTM

The first multimodal architecture (Figure 1a) takes as input fixed-size image and audio data. Our cross-connection design easily allows including residual cross-modal connections (adapted from the work of He et al. (2016)) that enable the raw input of one modality to directly interact with another modality's intermediate representation, potentially correcting any unwanted effects. Figure 1b illustrates both cross-modal and residual connections, which are constructed in a similar manner to

---

[*] www.cst.cam.ac.uk/~`{ccc53,pv273,pl219}`
[1] The dataset will be publicly released upon publication.

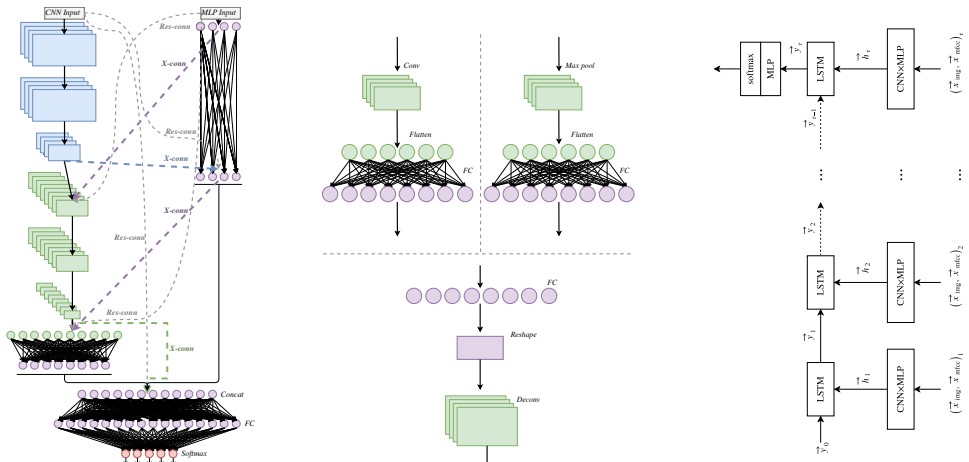

(a) CNN × MLP model with cross- and residual connections, denoted by X-conn (thick dashes) and Res-conn (thin dashes).

(b) *(Upper left:)* 2D⤳1D cross-connection. *(Upper right:)* 2D⤳1D residual connection. *(Bottom:)* 1D⤳2D cross-/residual connection.

(c) {CNN × MLP}–LSTM macro-scale: sequential processing across time steps. Input modalities are denoted by $\vec{x}_{\text{img}}$ and $\vec{x}_{\text{mfcc}}$, while $\vec{y}_t$ is the output of the LSTM layer at time $t$.

Figure 1: Overview of the two cross-modal architectures

the former. The second architecture (Figure 1c) has the advantage of not averaging the data across frames, thereby maintaining a richer source of features from both modalities. The feature extractor for a single frame is *weight-shared* across frames and corresponds to the model from Figure 1a without the last two layers. Additionally, all connections are designed as for the CNN × MLP architecture, but only operate within the single-frame feature extractor.

## 3 Experiments

### 3.1 Experimental Setup

We evaluated our models on the *AVletters* (Matthews et al., 2002) and *CUAVE* (Patterson et al., 2002) datasets. Both architectures were trained using the Adam SGD optimiser for 300 epochs, with hyperparameters as described by Kingma & Ba (2014) and a batch size of 128 for the CNN × MLP and 32 for the {CNN × MLP}–LSTM. The plots in Figures 2a and 2b show the evolution of validation accuracy and cross-entropy loss, respectively. A significant improvement over the baseline (same model, without cross-modal connections) can be seen in both plots.

For *AVletters*, the data was split into $k = 10$ folds—each fold corresponds to a *different person* in the dataset. For *CUAVE*, we used the pre-processing described by Ngiam et al. (2011) and $k = 9$. Additionally, we have curated a new dataset *Digits* showing 15 people saying the digits 0–9 in a *low*, *normal* and *loud* voice, *slowly* and *quickly*. We extracted two modalities: *image data* (2D video frames) and *audio data* (either 1D mel-frequency cepstral coefficients or 2D spectrograms).

### 3.2 Results

The results for all classifiers are shown in Table 1, with $p$-values indicating that the XFlow models improve on their respective baselines with statistical significance. The most impressive overall result was achieved by the {CNN × MLP}–LSTM architecture, showing remarkable benefits to temporal sequence modelling, as the corresponding baseline performed better than the CNN × MLP while using more than 5 times fewer parameters. Influenced by the previous results, we only used the recurrent model for comparison against *CorrRNN* (Yang et al., 2017), the latest published state-of-the-art result on *AVletters* and *CUAVE* (to the best of our knowledge), and the same train/test partition (Ngiam et al., 2011) seen in all previous published approaches. Both the baseline and

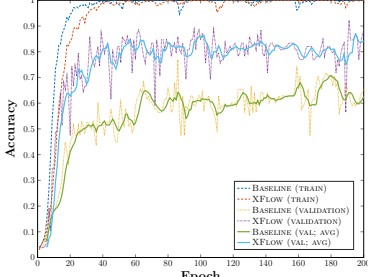 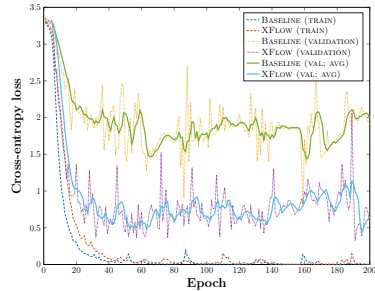

(a) Plot of the accuracy of the {CNN × MLP}–LSTM model on the first *AVletters* cross-validation fold.

(b) Plot of the cross-entropy loss of the {CNN × MLP}–LSTM model on the same fold.

Figure 2: Model optimisation, with a sliding averaging window of 5 epochs on the accuracy values, to emphasise the model capabilities during training.

XFlow models outperform *CorrRNN*, with a relative error improvement of 67.5% on *AVletters* and of 24.4% on *CUAVE*.

Table 1: $k$-fold cross-validation and comparative evaluation against the state-of-the-art approach. We used the paired $t$-test across fold results, with a significance threshold of $p \leq 0.05$.

| | AVletters | | | | Digits | | | CUAVE | | | |
| --- | --- | --- | --- | --- | --- | --- | --- | --- | --- | --- | --- |
| | Baseline | XFlow | CorrRNN | $p$-value | Baseline | XFlow | $p$-value | Baseline | XFlow | CorrRNN | $p$-value |
| CNN × MLP | 73.1% | **74.0%** | – | 0.65 | 78.3% | **86.7%** | $2 \times 10^{-3}$ | 90.3% | **93.5%** | – | 0.05 |
| {CNN × MLP}–LSTM (CV) | 78.1% | **85.6%** | – | 0.02 | 88.7% | **93.0%** | $1.2 \times 10^{-3}$ | 96.9% | **98.8%** | – | 0.01 |
| {CNN × MLP}–LSTM (holdout) | 91.5% | **94.6%** | 83.4% | – | – | – | – | 96.1% | **96.9%** | 95.9% | – |

## 3.3 Interpretability of Cross-connections

The visualisations in Figure 3 prove that each modality can be helpfully converted to the other one. In the 2D-1D direction, the outputs visibly produce clustering according to the classes (Figure 3a), whereas the 1D-2D transformation preserves dynamics across time steps for an entire video sequence (Figure 3b), presenting these dynamics in a structured 2D manner (Figure 3c). While these analyses represent a small step towards the general problem of neural network interpretability (Lipton, 2016), the results observed are largely encouraging. In particular, residual cross-connections pave the way for a methodology that lets us almost directly assess the way in which raw inputs of one kind relate to higher-level features of another kind, potentially allowing us to draw useful conclusions about cross-modal systems in general.

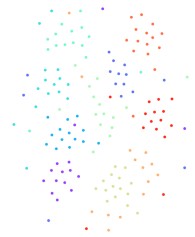 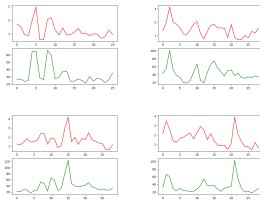 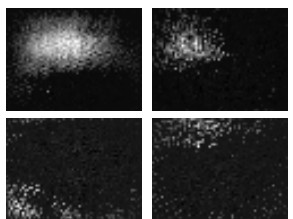

(a) Best viewed in colour. Two-dimensional $t$-SNE plot of the outputs of the second 2D⇝1D cross-connection within the CNN × MLP model. Each colour corresponds to a different class from the *Digits* dataset.

(b) Differences for 4 kernels in the final layer of the first {CNN × MLP}–LSTM residual connection. Horizontal axis: time, vertical axis: $L^2$ norm. *(Top:)* Differences between the 2D outputs. *(Bottom:)* Corresponding differences for the 1D inputs.

(c) Example outputs of the first {CNN × MLP}–LSTM residual connection corresponding to the same kernels as in Figure 3b.

Figure 3: Interpretability of cross-modal transformations (pre-trained on *Digits*)

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

## A    MODEL ARCHITECTURES

Tables 4a and 4b summarise the two models in terms of the number of parameters and cross-connections (residual connections can be then inferred from the shape of their target). All convolutional and fully-connected layers in the architectures have *ReLU* activations. In the CNN × MLP model, *batch normalisation* (Ioffe & Szegedy, 2015) is applied after the input, convolutional, first fully-connected (MLP stream) and the merge layers. We also applied *dropout* (Srivastava et al., 2014) with $p = 0.25$ after every max-pooling layer and with $p = 0.5$ after the first fully-connected (MLP stream), merge and final fully-connected layers. The {CNN × MLP}–LSTM model only employs batch normalisation after the input layer and merge layer, followed in the latter case by dropout ($p = 0.5$). We have taken steps to ensure integrity of the information and used the more general *PReLU* activation function (He et al., 2015) inside cross-modal connections.

Figure 4: Description of architectures (baselines and models with cross-connections, whose parameters are described in **bold**).

(a) CNN × MLP

| Output size | CNN stream | MLP stream |
|---|---|---|
| ([80 × 60, 16], 128) | [3 × 3, 16] Conv × 2 | Fully-connected 128-D |
| ([40 × 30, 16], 128) | 2 × 2 Max-Pool, stride 2 | |
| **([40 × 30, 32], 192)** | **[1 × 1, 16] Conv** **Fully-connected 64-D** ↘ | **Fully-connected 759-D** ↗ **[8 × 8, 16] Deconv** |
| ([40 × 30, 32], 128) | [3 × 3, 32] Conv × 2 | Fully-connected 128-D |
| ([20 × 15, 32], 128) | 2 × 2 Max-Pool, stride 2 | |
| **([20 × 15, 64], 256)** | **[1 × 1, 32] Conv** **Fully-connected 128-D** ↘ | **Fully-connected 204-D** ↗ **[4 × 4, 32] Deconv** |
| (256, 128) | Fully-connected 256-D | |
| 512 | Fully-connected 512-D | |
| | 26-way softmax | |

(b) {CNN × MLP}–LSTM

| Output size | CNN stream | MLP stream |
|---|---|---|
| ([80 × 60, 8], 32) | [3 × 3, 8] Conv | Fully-connected 32-D |
| ([40 × 30, 8], 32) | 2 × 2 Max-Pool, stride 2 | |
| **([40 × 30, 8], 64)** | **[1 × 1, 8] Conv** **Fully-connected 32-D** ↘ | **Fully-connected 375-D** ↗ **[16 × 16, 8] Deconv** |
| ([40 × 30, 16], 32) | [3 × 3, 16] Conv | Fully-connected 32-D |
| ([20 × 15, 16], 32) | 2 × 2 Max-Pool, stride 2 | |
| **([20 × 15, 64], 96)** | **[1 × 1, 16] Conv** **Fully-connected 64-D** ↘ | **Fully-connected 104-D** ↗ **[8 × 8, 16] Deconv** |
| (64, 32) | Fully-connected 64-D | |
| 64 | LSTM | |
| | 26-way softmax | |

Some of the architectures we developed contain a large number of parameters (underlined in Table 2 [2]). In such circumstances, initialisation heavily influences the representation learned during training in low-data scenarios (*AVletters* and *Digits*, which had approximately half as much training data as *CUAVE*). Consequently, the accuracies obtained might not always reflect the best performance that the classifier has the potential to obtain on a particular validation fold. Because of this, we trained each model underlined in Table 2 *five* times per validation fold, for the two datasets, and recorded the maximum result. The final accuracy for an architecture was then computed as the average over all folds. For statistical significance testing, we used the paired $t$-test across fold results, with a significance threshold of $p \leq 0.05$.

Table 2: Number of trainable parameters in each considered model, for the *AVletters* and *Digits/CUAVE* datasets.

| | **AVletters** | **Digits/CUAVE** |
|---|---|---|
| **Baseline** | | |
| CNN × MLP | 2,740,512 | 5,664,054 |
| {CNN × MLP}–LSTM | 353,650 | 721,170 |
| **XFlow** | | |
| CNN × MLP | 8,852,962 | 17,764,002 |
| {CNN × MLP}–LSTM | 1,387,488 | 2,967,388 |

---

[2]Depending on the dataset, the cross-connected models differ in the number of parameters. This is due to the cross-connections encompassing operations such as transposed convolution, which requires its output data to be in a specific shape for concatenation with the other stream.

