# OpenReview forum: "XFlow: Cross-modal Dataflow Neural Networks for Audiovisual Classification"
_ICLR.cc/2018/Workshop — Reject_

### Official Review · AnonReviewer2 · 2018-03-08
**Promising direction but not clear**

**Rating:** 4
**Confidence:** 4

**Review:**

Summary:
This paper proposes some ways to mixing deep representation from two modalities: audio and visual inputs. They evaluated on AVLetters and CUAVE datasets and seems to beat the baseline architecture without the proposed connections.

Strength:
- Cross-modal architecture design is a very important topic. We need more insights on how to mix representation from two difference sources. This paper is a great piece of effort towards this direction. Experimental results are also convincing.

Weakness:
- The paper is very unclear. I don’t know the exact formulas for X-conn and Res-conn. The only description of the model is to read the figure, which is not very intuitive. The connections in Figure 1 a) is hard to see. It seems to me that X-conn is just a reshape operator by reading Figure 1 b) and I still don’t know what is the exact form of ResConn.

- The experiments lack proper ablation studies. It seems that the baseline without any additional connections is obviously worse, but there are so many connections that are added. Which one is the most useful? What kind of effect does ResConn and X-Conn bring, separately?

- The section 3.3 is very confusing. I don’t understand the purpose of the plots, especially Figure 2 b) and c). It’s not clear what is “good” in those plots and what we are expected to see or learn about.

Conclusion:
Although the core theme of the paper is a promising direction, I feel that the current presentation of the paper does not convey clear methodology and experimentation to the readers. Therefore I recommend reject.

---

### Official Review · AnonReviewer1 · 2018-03-09
**Limited empirical evaluation**

**Rating:** 4
**Confidence:** 3

**Review:**

This paper investigates multimodal learning (image/video + sound) and investigate the use of cross-modality connection. Authors compare model with cross-connection versus baseline without cross connection on 3 different datasets (one of the dataset is a contribution of the paper as well). Empirical results show that cross-model achieve better results than previous works and baseline without the cross-correlation.

Proposed approach has somewhat limited novelty and the empirical evaluation is carried out on relatively small scale dataset.  I would recommend to evaluate the approach on larger dataset to better assess the benefit of  the approach. In addition it would be nice to add an ablation  to see the impact of the different individual cross connections.

---

### Official Review · AnonReviewer3 · 2018-03-11
**Good results but more analysis needed**

**Rating:** 6
**Confidence:** 5

**Review:**

Typical neural nets applied to classification tasks for multi-modal data consist of separate uni-modal pathways which are fused at the higher levels. The proposed model adds some extra connections from the lower layers of the unimodal pathways to the higher layers of the other modality's pathway. This architectural change is shown to lead to better classification performance.

This paper re-affirms the idea that skip connections help, which is somewhat widely known now. Unless there is some specific insight to be had, this particular instantiation of the idea would only be of moderate interest. The authors provide some analysis (section 3.3) but it is not immediately clear from that what is the point being made. However, the quantitative results show significantly better performance.

Minor corrections and suggestions-
- The terms "residual connection" and "cross connection" should be explained more clearly. This nomenclature seems confusing because both connections cut cross modalities and can be seen to be "residual". Based on Figure 1(b) the only difference between them seems to be that the first operation is conv vs max pool.

- In the abstract :"for performing cross-modality" : something missing here?

---

### Decision · Program_Chairs · 2018-03-20
**ICLR 2018 Workshop Acceptance Decision**

**Decision:**

Reject

**Comment:**

Based on the reviews, this paper has not been accepted for presentation at the ICLR workshop. However, the conversation and updates can continue to appear here on OpenReview.